# Effects of Age and Lifelong Moderate-Intensity Exercise Training on Rats’ Testicular Function

**DOI:** 10.3390/ijms231911619

**Published:** 2022-10-01

**Authors:** Joana V. Silva, Joana Santiago, Bárbara Matos, Magda C. Henriques, Daniela Patrício, Ana D. Martins, José A. Duarte, Rita Ferreira, Marco G. Alves, Paula Oliveira, Pedro F. Oliveira, Margarida Fardilha

**Affiliations:** 1Department of Medical Sciences, Institute of Biomedicine—iBiMED, University of Aveiro, 3810-193 Aveiro, Portugal; 2QOPNA & LAQV, Department of Chemistry, University of Aveiro, 3810-193 Aveiro, Portugal; 3Unit for Multidisciplinary Research in Biomedicine (UMIB), Institute of Biomedical Sciences Abel Salazar (ICBAS), University of Porto, 4050-313 Porto, Portugal; 4Research Center in Physical Activity, Health and Leisure (CIAFEL), Faculty of Sport, University of Porto, 4200-450 Porto, Portugal; 5Centre for Research and Technology of Agro-Environmental and Biological Sciences (CITAB), Inov4Agro, University of Trás-os-Montes and Alto Douro (UTAD), Quinta de Prados, 5000-801 Vila Real, Portugal

**Keywords:** aging, physical exercise, testicular atrophy, mitochondrial function, protein synthesis, stress response, rodent model

## Abstract

Aging is associated with testicular morphological and functional alterations, but the underlying molecular mechanisms and the impact of physical exercise are poorly understood. In this study, we examined the effects of age and lifelong moderate-intensity exercise on rat testis. Mature adults (35 weeks) and middle-aged (61 weeks) Wistar Unilever male rats were maintained as sedentary or subjected to a lifelong moderate-intensity treadmill training protocol. Testis weight and histology, mitochondrial biogenesis and function, and proteins involved in protein synthesis and stress response were evaluated. Our results illustrate an age-induced testicular atrophy that was associated with alterations in stress response, and mitochondrial biogenesis and function. Aging was associated with increased testicular levels of heat shock protein beta-1 (HSP27) and antioxidant enzymes. Aging was also associated with decreased mRNA abundance of the nuclear respiratory factor 1 (*Nrf1*), a key transcription factor for mitochondrial biogenesis, which was accompanied by decreased protein levels of the oxidative phosphorylation system (OXPHOS) complexes subunits in the testes of older animals. On the other hand, exercise did not protect against age-induced testicular atrophy and led to deleterious effects on sperm morphology. Exercise led to an even more pronounced decrease in the *Nrf1* mRNA levels in testes of both age groups and was associated with decreased mRNA abundance of other mitochondrial biogenesis markers and decreased protein levels of OXPHOS complexes subunits. Lifelong moderate-intensity exercise training was also associated with an increase in testicular oxidative stress markers and possibly with reduced translation. Together, our results indicate that exercise did not protect against age-induced testicular atrophy and was not associated with beneficial changes in mitochondria and stress response, further activating mechanisms of protein synthesis inhibition.

## 1. Introduction

Infertility affects 15–20% of couples in reproductive age; and, in half of the cases, the cause is associated with male dysfunctions [1]. Approximately 40% of the male infertility cases are idiopathic and it has been suggested that part of those cases arise from advanced paternal age [2,3,4]. Several studies have reported the detrimental effects of aging on semen parameters and sperm DNA integrity; however, literature on the impact of aging on testicular function, particularly at the cellular and molecular levels, has been limited and controversial [5]. Lifestyle behaviors, including sedentary lifestyle and obesity, have also been implicated in male infertility. There is increasing knowledge on the health benefits of exercise; however, there is evidence that high-intensity exercise may compromise testicular function, resulting in a decrease in the levels of total and free testosterone, luteinizing and follicle-stimulating hormones and a deterioration in sperm parameters [6]. This might be explained by elevated scrotal temperatures associated with some sports and alterations in the secretion of gonadotropins and androgens [7]. Still, most studies on this topic focus on the association between exercise and seminal quality. Some have reported deleterious effects of exercise training on seminal parameters such as sperm concentration, which is a reflection of testicular function [8,9,10]. Others, however, have reported that seminal parameters improved with exercise training [11,12] or stated that physical activity is not related to semen quality [13,14]. Conclusions are difficult to draw due to the lack of standardization in research protocols, especially concerning exercise type, intensity and volume, and study population. Most studies have focused on the effects of high-intensity training among athletes [9,15,16] or subfertile/infertile men [8,17], which makes it difficult to extrapolate to the general population. Additionally, it is challenging to exclude confounding factors such as exposure to endocrine disrupting chemicals, diet quality, stress and other situations associated with elevated scrotal temperatures. 

Since most studies on the effects of aging and exercise focus on seminal quality rather than on testicular features, and to eliminate the confounding factors, in this study we examined the effects of age and lifelong moderate-intensity exercise on testicular structural and molecular characteristics in rats.

## 2. Results

### 2.1. Age and Exercise Training Impact Rats’ Body and Testes Weight

Rats’ body weight, testis weight and testis mass-to-body weight ratio of the different experimental groups are listed in Table 1. There was an increase in body weight in the groups of older rats when compared with the groups of younger rats (in the sedentary groups, mean increase of 15.2%; in the exercised groups, mean increase of 9.7%). Concerning the testis mass-to-body weight ratio, there was a decrease in the group of older sedentary rats when compared with the younger sedentary rats (mean decrease of 17.9%).

A decrease in body weight in both exercised groups in comparison with the sedentary animals was observed (at 35 W mean decrease of 14.3%; at 61 W mean decrease of 18.4%). There was also a decrease in testis mass-to-body weight ratio in the exercised animals in comparison with the sedentary groups (at 35 W mean decrease of 26.5%; at 61 W mean decrease of 4.9%). Older animals subjected to exercise training presented significantly higher levels of serum testosterone compared with sedentary animals with the same age and exercised younger animals.

### 2.2. Aging Decreases Testis Cellular Density and Exercise Decreases Rat’s Seminiferous Tubules Area and Increases Basal Lamina Thickness

There were no significant differences in the number of seminiferous tubules per microscopic field in any of the study groups (Figure 1b,c; Appendix A). The groups of older rats exhibited a decreased cellular density with few spermatids and mature spermatozoa in the center of seminiferous tubules, compared with the group of younger rats (Figure 1b). 

In the exercised animals, there was a decrease in seminiferous tubules area when compared with the sedentary groups (at 35 W mean decrease of 22.2%; at 61 W mean decrease of 35.8%) (Figure 1b,c; Appendix A). Concerning the basal lamina thickness, the younger exercised group revealed an increase when compared with the sedentary animals of the same age (mean increase of 11.7%) (Figure 1b,d; Appendix A). The older exercised animals, in addition to the decreased cellular density, presented immature germ cells at the center of the seminiferous tubules. In fact, the older exercised animals presented a decrease in sperm concentration and in the percentage of morphologically normal sperm (Table 2). Exercise was associated with an increase in the percentage of decapitated head (DH) and tail defects (TD) (Table 2).

### 2.3. Age and Exercise Compromise Rat’s Testicular Mitochondrial Biogenesis

We evaluated the expression of key genes (*Sirt1*, *Pgc1a* and *Nrf1*) involved in the regulation of mitochondria dynamics to assess the impact of age and exercise on testicular function (Figure 2). Pgc1α, the master player of mitochondrial biogenesis, controls the expression of genes involved in energy homeostasis, mitochondrial biogenesis, fatty acid oxidation and glucose metabolism [18]. Sirt1 converts inactivated Pgc1α to the active form and active Pgc1α stimulates the expression of *Nrf1* and *Nrf2* that act on the nuclear genes coding for subunits of the OXPHOS system [19,20]. A decrease in nuclear respiratory factor 1 (*Nrf1)*-relative mRNA abundance was observed in the group of older sedentary rats in comparison with the group of younger sedentary animals (mean decrease of 21.8%) (Figure 2a). No age-associated significant alteration was observed in *Nrf1*-relative mRNA abundance in the exercised groups. 

There was a decrease in the exercised groups in comparison with the non-exercised groups in the relative mRNA abundance of *Nrf1* (at 35 W mean decrease of 57.9%; at 61 W mean decrease of 38.8%), peroxisome proliferator-activated receptor gamma coactivator 1α *(Pgc1a)* (at 35 W mean decrease of 74.3%; at 61 W mean decrease of 34.1%) and Sirtuin 1 (*Sirt1)* (at 35 W mean decrease of 48.6%; at 61 W mean decrease of 24.4%) (Figure 2a–c).

### 2.4. Age Increases Testicular mtDNA Copy Number in Rat Testis

The mtDNA copy number was also evaluated by quantifying the number of MT-ND1 copies, which codifies for a subunit of the protein NADH dehydrogenase 1. This protein is part of mitochondrial complexes, specifically complex I that is responsible for the first step in the electron transport. The mtDNA copy number was increased in group of older sedentary animals, in relation to the levels detected in the younger sedentary group (mean increase of 53.0%) (Figure 2d). No age-associated significant alteration was observed in mtDNA copy number in the exercised groups. 

### 2.5. Age and Exercise Decrease OXPHOS Complexes II, III and V Expression in Rat Testis

Mitochondria are responsible for adenosine triphosphate (ATP) synthesis by oxidative phosphorylation. The electrons resulting from this process are guided through redox carriers (mitochondrial complexes I, II, III, and IV) and linked with proton pumping by an F-ATPase (mitochondrial complex V).

A decrease in mitochondrial complex II expression was detected in the groups of older rats when compared with groups of younger rats (in the sedentary groups, mean decrease of 70.9%; in the exercised groups, a mean decrease of 84.9%) (Figure 2e). A decrease in the older non-exercised animals in comparison with the younger non-exercised rats was detected in the expression of mitochondrial complexes III (mean decrease of 62.8%) and V (mean decrease of 41.6%) (Figure 2f,g). No age-associated significant alteration was observed in the expression of mitochondrial complexes III and V in the exercised groups.

A reduction in the levels of complexes III (mean decrease of 61.4%) and V (mean decrease of 36.0%) was observed in the exercised younger animals when compared with their age-matched non-exercised animals (Figure 2f,g). A decrease in complex II was observed in the older exercised group in comparison with non-exercised rats of the same age (mean decrease of 48.4%) (Figure 2e).

No significant alterations in the expression of the mitochondrial complex I were observed in any of the experimental groups (Appendix A). 

### 2.6. Age Increases HSP27 Phosphorylation and Antioxidant Enzymes Expression

In the groups of older animals there was an increase in the protein levels of heat shock protein beta-1 (HSP27) (in the sedentary groups, mean increase of 20.4%; in the exercised groups, a mean increase of 98.2%) and phosphorylated-HSP27 (in the sedentary groups, mean increase of 18.7%; in the exercised groups, a mean increase of 7.3%) (Figure 3a,b). There was also an increase in the groups of older animals, in the levels of the antioxidant enzymes phospholipid hydroperoxide glutathione peroxidase (GPx4) (in the sedentary groups, mean increase of 37.0%; in the exercised groups, a mean increase of 61.0%), superoxide dismutase [Cu-Zn] (SOD1) (in the sedentary groups, mean increase of 27.8%; in the exercised groups, a mean increase of 32.2%) and mitochondria SOD (SOD2) (in the sedentary groups, mean increase of 25.0%; in the exercised groups, a mean increase of 47.4%) (Figure 3c–e). No significant alterations were observed in the levels of cleaved-Caspase-3 in any experimental group (Appendix A).

### 2.7. Exercise Potentially Attenuates Protein Synthesis and Increases Oxidative Stress in Rat Testis

The levels of phosphorylated ribosomal protein S6 kinase beta-1 (S6K1, also known as p70S6 kinase 1), a major mammalian target of rapamycin complex 1 (mTORC1) substrate whose phosphorylation promotes protein synthesis and cell proliferation, were decreased in the exercised groups in comparison with the sedentary animals (at 35 W mean decrease of 23.7%; at 61 W mean decrease of 34.2%) (Figure 3f). No age-associated differences were observed. 

The levels of phosphorylated eukaryotic translation initiation factor 2 subunit 1 (eIF2α) were increased in the younger exercised animals when compared with the sedentary group (mean increase of 36.3%) (Figure 3g). Phosphorylation of eIF2α is a well-documented mechanism to downregulate protein synthesis under a variety of stress conditions [21,22]. No significant alterations were observed in the total levels of eIF2α (Appendix A).

Protein phosphatase 1 (PP1) regulatory subunit 15A (PPP1R15A, also known as GADD34) recruits PP1 to dephosphorylate eIF2α, thereby reversing the shut-off of protein synthesis initiated by stress-inducible kinases and facilitating recovery of cells from stress [21,23]. The levels of GADD34 were increased in the exercised groups in comparison with the non-exercised animals (at 35 W mean increase of 10.7%; at 61 W mean increase of 67.2%) (Figure 3h). On the other hand, the levels of PP1γ2, the testis-enriched and sperm-specific PP1 isoform, were decreased in the exercised animals (at 35 W mean decrease of 16.5%; at 61 W mean decrease of 26.8%) (Figure 3i).

There was an increase in the levels of the small heat shock protein HSP27 in exercised animals in comparison with the sedentary groups (at 35 W mean increase of 32.6%; at 61 W mean increase of 118.3%) (Figure 3a). There was also an increase in the levels of the antioxidant enzyme SOD1 in the exercised groups (at 35 W mean increase of 47.6%; at 61 W mean increase of 52.6%) (Figure 3d) and SOD2 in the older exercised group compared with the younger group (at 61 W mean increase of 25.9%) (Figure 3e).

## 3. Discussion

There are many discrepancies in studies regarding the effect of aging and exercise on male fertility. Additionally, most studies on the effects of aging and exercise focus on seminal quality, with very few focusing on testicular features. Therefore, the aim of the present study was to add new insights on the impact of age and lifelong moderate-intensity exercise training on testicular structure and function using a rat model. When interpreting these results, it is important to consider that testes are a highly heterogenous type of tissue, composed of many types of cells (Sertoli and Leydig cells, germ cells—spermatogonia, spermatocyte, spermatid, spermatozoa, myoid cells, etc.), with different dependence on mitochondria activity. Therefore, it will be important in the future to characterize the different types of cells both in terms of mitochondrial biogenesis and antioxidant profile, to further deepen the evidence here presented. 

Several studies reported that as men age increases, testosterone levels decrease (reviewed by [5]). On the other hand, moderate-intensity training seems to improve hormone serum levels, including total and free testosterone and sex hormone binding globulin, despite contradicting results exist (reviewed by [6]). We showed that serum testosterone levels did not change significantly with age or after physical exercise in the younger rats except in the group of exercised older rats in which the levels of testosterone were significantly higher than in the other groups. We observed that the groups of older animals exhibited an increase in body weight and testicular atrophy, which was previously reported in several studies (reviewed by [5]). Testicular atrophy can be, in part, explained by impaired mitochondrial function. Numerous studies support a direct link between aging and mitochondrial function decline [24,25]. We observed that age was associated with decreased *Nrf1* mRNA abundance, a key transcription factor for mitochondrial biogenesis that activates the nuclear genes coding for subunits of the oxidative phosphorylation system (OXPHOS) [19], which was further confirmed by decreased protein levels of the OXPHOS complexes subunits in the groups of older rats. Joseph et al. previously reported an age-induced atrophy in rat testes [26]. Still, they did not observe age-related alteration neither in key mitochondrial regulatory proteins—including *Pgc1a* and *Nrf1*—or in apoptotic susceptibility. Nonetheless, they still speculated mitochondrial function may be impaired with age and contribute to testicular atrophy [26], which we reported in this study. *Nrf1* and Sirt1 also seems to regulate testosterone biosynthesis [27,28,29], which could partially explain some of the alterations in testosterone levels observed.

We observed an increased mtDNA copy number in the group of older sedentary animals in comparison with the younger sedentary group. This was not accompanied by an increase in the expression of OXPHOS complexes. In fact, a decrease in components of the mitochondrial electron transport chain (complex II and III) and ATP synthase (complex V) was observed in the older sedentary group. Additionally, no alterations in the levels of *Pgc1a* were observed in those animals. Thus, our data indicate age-induced decrements in mitochondrial key regulatory players but increased mtDNA copy numbers, which may be due to an age-related deficient removal of dysfunctional mitochondria and damaged mtDNA [30,31]. In fact, reduced autophagy associated with aging facilitates the accumulation of damaged mitochondria and oxidized proteins, resulting in high levels of reactive oxygen species (ROS) production. Several molecular mechanisms of aging have been proposed, including increase in oxidative stress and free radical production, due to the alterations in the enzymatic activity of antioxidant molecules, especially in Leydig cells, usually resulting in reduced testosterone levels [32,33,34,35]. To avoid harmful effects from ROS, the testes mainly use the enzymatic copper/zinc SOD (SOD1), mitochondrial SOD (SOD2), extracellular SOD (SOD3), catalase (CAT), and selenoenzyme phospholipid hydroperoxide glutathione peroxidase (GPx4) [36]. However, the relationship between aging and antioxidant system remains controversial: while some studies reported a decreased activity of antioxidant enzymes with age [37], others showed that aging was not linked to a decline in the levels of these enzymes but, contrarily, to an increase [38]. In our study, the levels of the antioxidant enzymes GPx4, SOD1 and SOD2 were increased in the groups of older rats, consistent with some other reports [38]. Additionally, the levels of total and phosphorylated-HSP27, increased in the groups of older animals. Cumulative evidences support the role of HSP27 in inflammation and cellular protection from cytokines and ROS [39,40,41]. Following cellular stress, these small HSPs are expressed and form oligomeric structures. The role of phosphorylation in regulating HSP27 chaperone activity is controversial. Some studies report that phosphorylation dissociates HSP27 from large small heat-shock protein oligomers which impairs its chaperone activity and ability to protect cells against stress [42,43]; while others found that phosphorylated HSP27 protects better than does the unphosphorylated form [44]. However, studies investigating the age impact on the phosphorylation state of HSP27 are still missing. Our results support the hypothesis that aging induces stress in testis, leading to an increased production of ROS with a consequent increase in molecular chaperones such as HSP27, and antioxidative enzymes, such as SOD1 and GPx4.

Concerning the exercised groups, as expected, a decrease in body weight in comparison with their age-matched non-exercised groups was observed. A decrease in testis weight in the exercised groups was also observed. In fact, exercise induced a decrease in seminiferous tubules area in both age groups and a thickening of the basal lamina in the group of younger rats. Exercise-induced mitochondrial adaptations have also been widely investigated [45]. In this study, exercise led to an even more pronounced decrease in the *Nrf1* mRNA levels in both age groups and was also associated with decreased mRNA abundance of other mitochondrial biogenesis markers—*Sirt1* and *Pgc1a*—and decreased protein levels of the OXPHOS complexes subunits. Joseph et al. also investigated mitochondrial adaptations induced by exercise (10 weeks treadmill training) in testis of middle aged (6 months) and old (24 months) rats [26]. They observed that exercise training started in late-life either protected against age-induced testicular atrophy, and mitochondrial adaptations appear to partially contribute toward this improvement observed in the old rats [26].

To further elucidate the impact of lifelong exercise in rat testes, proteins involved in key cellular processes, such as stress response and protein synthesis were evaluated. As previously showed, older exercised animals presented significantly higher levels of serum testosterone compared with sedentary animals accompanied by an increase in the levels of the antioxidant enzymes SOD1 and SOD2 and higher levels of the molecular chaperone HSP27, suggesting that exercise induces stress in the testis. Many reports showed that different types of physical activity, including exercise training, induce the production of ROS in several types of cells [46,47] and upregulate antioxidant activity in rat testis [48,49]. We also showed that exercise induced beneficial mitochondrial adaptations especially in old animals including increase in mitochondrial antioxidant capacity (SOD2), despite mitochondrial biogenesis and activity decrease, as previously discussed. This is in part consistent with other studies that reported mitochondria-mediated ROS overload after high-intensity continuous running [50]. Testosterone seems to have important mitoprotective and antimitophagy functions in many tissues [51,52,53,54], alleviating mitochondrial ROS accumulation [55] and inducing the expression of *Nrf1* [53], which was not observed in this study. Despite quite contradictory, these results may suggest a possible mechanism of protection in testis in which aging and physical exercise impair mitochondrial activity leading to an increased production of ROS with a consequent activation of the antioxidant defense system to prevent severe testicular damage.

In this study, we also showed, for the first time, that the levels of p-p70S6 kinase 1 were significantly reduced in the exercised groups compared with sedentary animals, while no differences were observed with age. p70S6 kinase 1 is a major mTORC1 substrate, whose phosphorylation facilitates protein synthesis and promotes cell proliferation [56]. Previous studies showed that suppression of p70S6 kinase 1 phosphorylation results in inhibition of spermatogenesis by activation of autophagy in rat testes [57]. This may be a self-protective mechanism of the testes in response to an external stress [57]. The reduced levels of p70S6 kinase 1 phosphorylation suggest that the mTOR signaling is downregulated in testis of individuals subjected to exercise training. Reduced levels of p-p70S6 kinase 1 are also associated with protein synthesis repression [56]. Consistent with this evidence, we showed that the levels of p-eIF2α were increased in younger exercised animals compared with their age-matched sedentary group; and the levels of GADD34 and PP1γ2 were increased and decreased, respectively, in the exercised groups. Phosphorylation of eIF2α at Ser51 is a well-documented mechanism to decrease translation initiation, and thus downregulate protein synthesis under a variety of stress conditions [39]. GADD34 mediates eIF2α dephosphorylation though the binding and recruitment of PP1, restoring protein synthesis initiated by stress-inducible kinases and facilitating recovery of cells from stress [23,58,59]. These results suggest that exercise may be attenuating the overall protein synthesis, even by decreasing phosphorylated levels of p70S6 kinase 1, as by increasing eIF2α phosphorylation. The downregulation of the complex PP1/GADD34 due to reduced levels of PP1 in exercised groups also support these findings.

Overall, our results indicate that the age-induced atrophy in testes may be associated with alterations in mitochondrial biogenesis and function and stress response. Furthermore, exercise did not protect against age-induced testicular atrophy and was not associated with beneficial changes in mitochondria. In fact, lifelong moderate-intensity exercise training led to an upregulation of antioxidative enzymes and molecular chaperones, possibly activating mechanisms of protein synthesis inhibition.

## 4. Materials and Methods

### 4.1. Animals

Wistar Unilever male rats, aged four weeks old, were acquired from Charles River Laboratories (Lion, France). Animals were subjected to quarantine for two weeks and then randomly distributed into polycarbonate cages (five animals per cage). The cohort was then randomly divided into four experimental groups: Sedentary, 35 weeks (*n* = 8); Exercised, 35 weeks (*n* = 9); Sedentary, 61 weeks (*n* = 9); and Exercised, 61 weeks (*n* = 9) (Figure 1a). All animals were maintained in the animal facilities of University of Trás-os-Montes and Alto Douro (UTAD) according to the Directive 2010/63/EU recommendations, in controlled conditions, including a temperature of 22 ± 2 °C, a 12 h light/dark cycle and a relative humidity of 50 ± 10%. Animals were allowed to access food and water ad libitum (Mucedola 4RF21^®^, Milan, Italy). The experimental protocol was approved by the animal well-being responsible organ of UTAD and by Direção Geral de Alimentação e Veterinária-DGAV (license n° 021326).

### 4.2. Endurance Training Protocol

The animals of exercised groups started the endurance training protocol in a treadmill (Treadmill Control LE 8710, Harvard Apparatus, Holliston, MA, USA) at the age of eight weeks. The exercise program started with 30 min per day during the first week (habituation period) and was then increased to 60 min per day and maintained for 26 or 52 weeks (5 days per week) (Figure 1a). The speed of the treadmill was set to 70% of the maximal speed capacity of the animals and, every 15 days, the speed capacity was re-evaluated to correct for exercise intensity. To submit the sedentary animals to a similar stress, rats were regularly placed on a stationary treadmill for a few minutes.

### 4.3. Sample Collection

The animals were euthanized by an overdose of ketamine and xylazine (anesthetics), administered intraperitoneally, followed by exsanguination through cardiac puncture. The animals’ weight was measured and blood, testes and epidydimal sperm were collected. The testes were weighed and immediately processed for histological analysis or stored at −80 °C for further molecular analyses.

### 4.4. Determination of Serum Testosterone Levels

Serum testosterone levels were determined using a testosterone ELISA kit (Caymann Chemical, Ann Arbor, MI, USA) according to the manufacturers’ instructions Briefly, eight standards with decreasing testosterone concentrations were prepared, after which the absorbance at 412 nm was recorded using a microplate reader (Tecan^®^ Infinite M200) and a standard curve was generated. The serum samples were diluted and, based on the standard curve, the testosterone concentrations were determined in pg/mL.

### 4.5. Semen Analysis

Sperm concentration was determined using a Neubauer chamber. The Diff-Quik^TM^ rapid staining method was used to assess sperm morphology. Sperm cells were analyzed using a Zeiss optical Primo Star microscope (Carl Zeiss AG, Oberkochen, Germany).

### 4.6. Histological Analysis of Rat Testis

Testes were fixed in 4% paraformaldehyde and embedded in paraffin to form paraffin-blocks. The paraffin blocks were sectioned (5 µm) with a manual microtome and the sections were mounted in glass slides. Glass slides were deparaffinized in xylol, hydrated with alcohol in decreasing concentrations (100%, 95% and 75%) and stained either with haematoxylin and eosin (H&E) or Sirius red. The specimens were then examined in a bright-field optical microscope and digital images were captured, using ZEN Microscopy software (Zeiss, Oberkochen, Germany). These images were further analyzed using the ImageJ software [60] and the area and number of seminiferous tubules and basal lamina thickness were measured in three random microscopic fields.

### 4.7. Testis Preparation for DNA, RNA and Protein Extraction

The left testes were ground to powder using liquid nitrogen in a mortar and stored at −80 °C until use. The DNA specimens were prepared using the NZY Tissue gDNA Isolation Kit (NZYtech, Lisbon, Portugal). Extraction of RNA was performed using the GRS Total RNA (GRISP, Lda, Porto, Portugal) according to the manufacturer instructions. RNA concentrations were determined by NanoDrop™ 2000 Spectrophotometers (Thermo Fisher Scientific, Carlsbad, CA, USA). RNA was reversely transcribed using the NZY M-MuLV Reverse Transcriptase (NZYtech, Lisbon, Portugal). The resultant complementary DNA (cDNA) was used with exon-exon spanning primer sets designed to amplify cDNA fragments. For protein extraction, an adequate volume of ice-cold 1X radioimmunoprecipitation assay buffer (RIPA) (1 mL of buffer per 250 mg of testicular tissue) was added to the macerated tissue and incubated for 30 min at 4 °C with agitation. The lysates were centrifuged for 20 min at 4 °C at 16,000× *g* and the supernatant (soluble fraction) collected and stored at −30 °C. The Thermo Scientific™ Pierce™ BCA Protein Assay Kit (Fisher Scientific, Loures, Portugal) was used to determine the protein concentration of the samples according to manufacturer’s instructions.

### 4.8. Quantitative Reverse Transcriptase Polymerase Chain Reaction (qRT-PCR) Analysis of Mitochondrial Biogenesis

qRT-PCR was performed to evaluate the mRNA abundance of nuclear respiratory factor-1 (*Nrf1*), peroxisome proliferator-activated receptor gamma coactivator 1α (*Pgc1a*), Sirtuin 1 (*Sirt1*) and β2- macroglobulin (used to normalize gene expression levels). Fold variation of gene expression levels was calculated using the formula 2-ΔΔCt [56]. 

qRT-PCR was also used to determine the mtDNA copy number in rat testis. qRT-PCR was performed to evaluate the DNA abundance of mitochondrial NADH dehydrogenase 1 (*MT-ND1*), that was normalized with β2-microglobulin (single copy gene). The average of all three measurements was calculated. All the primer sets used in qRT-PCR experiments are presented in Appendix A.

### 4.9. Mitochondrial Respiratory Chain Complexes Detection by Immunoblot

Western blot was performed using standard methods. In brief, protein samples (30 µg) were fractionated on a 12% sodium dodecyl sulfate-polyacrylamide gel electrophoresis (SDS-PAGE) and transferred to polyvinylidene difluoride membranes. The membranes were blocked and incubated with primary antibodies (Total OXPHOS Rodent WB Antibody Cocktail, 1:1000, ab110413, Abcam, Cambridge, MA, USA) overnight at 4 °C. Mouse anti-GAPDH was used as the protein-loading control. The immune-reactive proteins were detected with secondary antibodies. Membranes were reacted with the ECF detection system (GE Healthcare, Chicago, IL, USA) and visualized with the BioRad GelDoc XR (Bio-Rad, Sintra, Portugal). The densities from each band were obtained using the Quantity One software (Bio-Rad, Sintra, Portugal).

### 4.10. Slot Blot

Protein samples were diluted to 0.002 µg/µL and blotted under vacuum into a nitrocellulose membrane, 0.45 µm pore size (GE Healthcare, Chicago, IL, USA) inside the slot blot device (BioRad Portugal, Sintra, Portugal). The membranes were blocked with 5% (w/v) Bovine Serum Albumin (BSA) in tris-buffered saline (TBS), for 1 h at room temperature (RT). Incubation with the primary antibodies occurred for 1 h at RT using the following dilutions: anti-p-p70S6 kinase 1 (Ser 434) (1:1000, sc-8416, Santa Cruz Biotechnology, Inc., Dallas, TX, USA), anti-p70S6 kinase 1 (1:1000, sc-393967, Santa Cruz Biotechnology, Inc., Dallas, TX, USA), anti-p-HSP27 (Ser82) (1:1000, sc-166693, Santa Cruz Biotechnology, Inc., Dallas, TX, USA), anti-HSP27 (1:1000, sc-13132, Santa Cruz Biotechnology, Inc., Dallas, TX, USA), anti-eIF2α (1:1000, 9722S, Cell Signaling Technology Europe Inc., Leiden, The Netherlands), anti-p-eIF2α (Ser 51) (1:1000, 119A11, Cell Signaling Technology Europe Inc., Leiden, The Netherlands), anti-PP1γ2 (1:5000, produced in house against the PP1γ2-specific C-terminal peptide—VGSGLNPSIQKASNYRNNTVLY [61]), anti-GADD34 (1:1000, sc-373815, Santa Cruz Biotechnology, Inc., Dallas, TX, USA), anti-GPx4 (1:1000, ABC269, Millipore, MA, USA), anti-SOD1 (1:1000, MABC684, Millipore, Massachusetts, EUA), anti-SOD2/MnSOD (1:1000, ab13533, Abcam, Cambridge, UK) anti-Caspase 3 (1:1000, GTX110543, GeneTex, Irvine, CA, USA), anti-Cleaved Caspase 3 (1:200, AB3623, Millipore, Burlington, Massachusetts, EUA). After being washed in TBS with 0.1% Tween™ 20 (TBS-T) (Fisher BioReagents™ by Thermo Fisher Scientific, Waltham, MA, USA), membranes were incubated with appropriate secondary antibodies, either IRDye^®^ 800CW goat anti-mouse (1:10,000) or IRDye 680RD^®^ goat anti-rabbit (1:10,000), for 1 h at RT. Membranes were scanned using the Odyssey Infrared Imaging System (LI-COR^®^ Biosciences, Lincoln, NE, USA). Results were normalized to Ponceau staining used before antibody probing. Phospho-proteins were normalized against the total level of the target protein.

### 4.11. Statistical Analysis

Descriptive statistics of all data were calculated. The statistical significance of the effects of age and lifelong moderate-intensity exercise on testicular structural and molecular characteristics in rats was calculated with Mann–Whitney U-tests. The significance level was set at 0.05. All analyses were conducted using Stata version 13.0 (StataCorp, College Station, TX, USA).

## Figures and Tables

**Figure 1 ijms-23-11619-f001:**
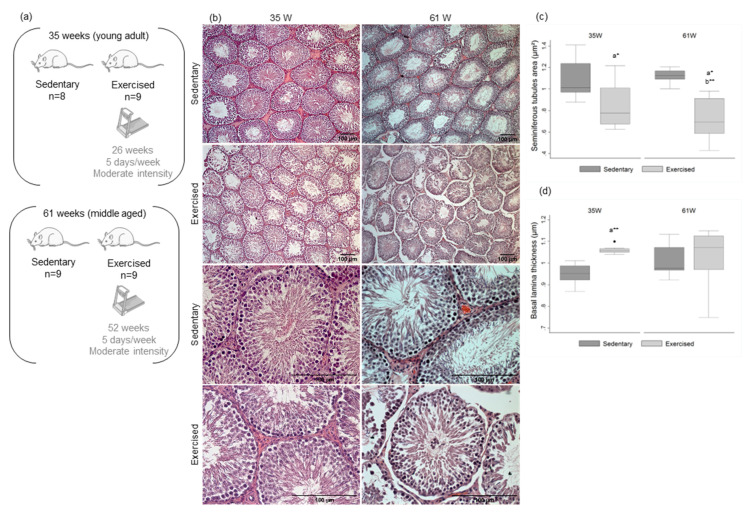
Experimental groups and effect of age and exercise training on rats’ testicular morphology. (**a**) Experimental groups. (**b**) Representative H&E stains of histological morphology of rat testes of the four experimental groups (upper panels, ×100 amplification; lower panels, ×400 amplification). (**c**,**d**) Box and whisker plots show the (**c**) area of seminiferous tubules and (**d**) basal lamina thickness of the four experimental groups. The horizontal line displays the median, the box edges show the 25th and 75th percentiles and the whiskers show the smallest and highest value within 1.5 box lengths from the box. (*n* = 8 for 35 W Sedentary; *n* = 9 for 35 W Exercised, 61 W Sedentary and 61 W Exercised). * *p* < 0.05; ** *p* < 0.01. a—Significantly different from 35 W Sedentary; b—significantly different from 61 W Sedentary.

**Figure 2 ijms-23-11619-f002:**
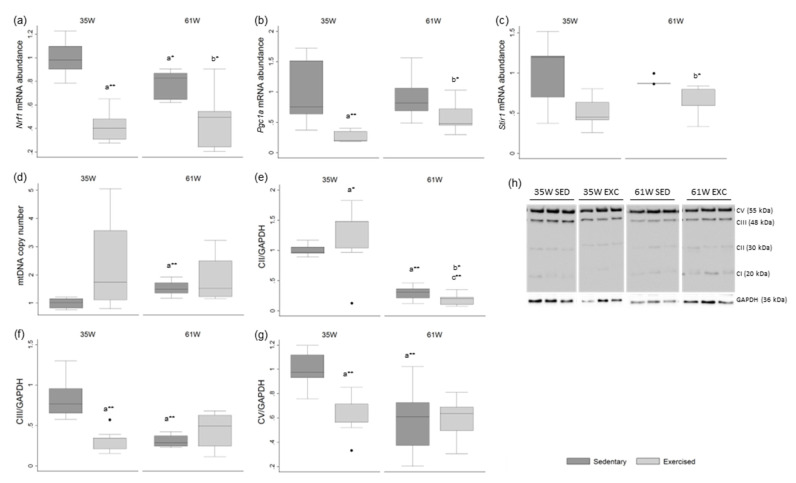
Box and whisker plots show the effect of age and exercise training on (**a**) nuclear respiratory factor 1 (*Nrf1*), (**b**) peroxisome proliferator-activated receptor gamma coactivator 1α (*Pgc1a*), (**c**) Sirtuin 1 (*Sirt1*) mRNA abundance in rat testis. Quantification of mtDNA abundance (**d**) in rat testis was also performed. The protein levels of mitochondrial respiratory chain complexes II, III and V are represented in (**e**–**g**), respectively. (**h**) Representative Western blot images of mitochondrial respiratory chain complexes II, III and V and GAPDH. CI: NADH dehydrogenase (ubiquinone), 1 beta subcomplex, subunit 8; CII: succinate dehydrogenase complex, subunit B, iron sulfur; CIII: ubiquinol-cytochrome c reductase, core protein II; CV: ATP synthase alpha-subunit. GAPDH was used as the protein-loading control and the results are represented in fold variation to control. The horizontal line displays the median, the box edges show the 25th and 75th percentiles and the whiskers show the smallest and highest value within 1.5 box lengths from the box (*n* = 8 for 35 W Sedentary; *n* = 9 for 35 W Exercised, 61 W Sedentary and 61 W Exercised). * *p* < 0.05; ** *p* < 0.01. a—Significantly different from 35 W Sedentary; b—significantly different from 61 W Sedentary; c—significantly different from 35 W Exercised.

**Figure 3 ijms-23-11619-f003:**
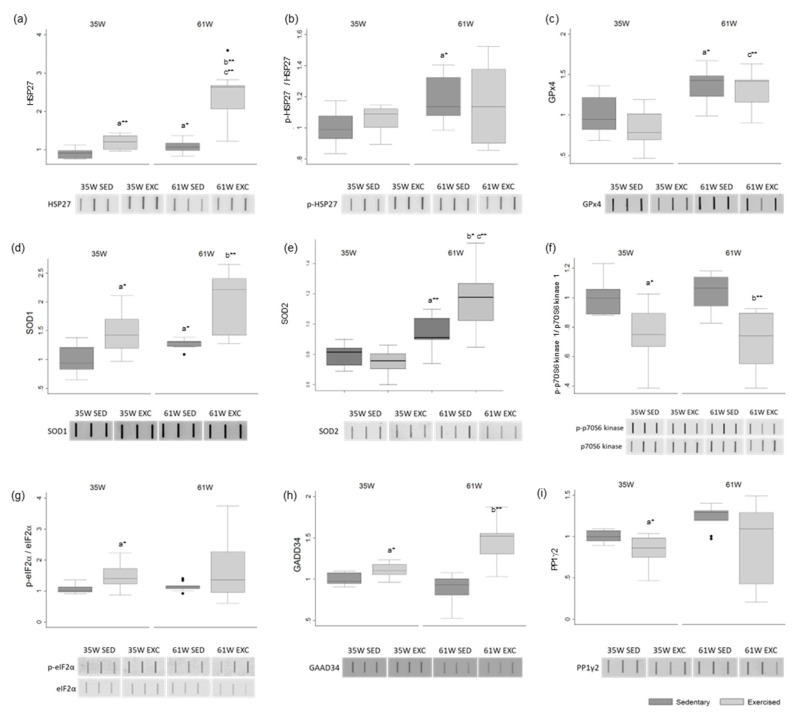
Box and whisker plots show the effect of exercise training and age on the protein levels of (**a**) HSP27, (**b**) p-HSP27, (**c**) GPx4, (**d**) SOD1, (**e**) SOD2, (**f**) p-RPS6KB1, (**g**) p-eIF2a, (**h**) GADD34 and (**i**) PP1g2. Ponceau S. was used as the protein-loading control. Phosphorylation-specific signals were normalized against the total level of the total protein. The results are represented in fold variation to control. The horizontal line displays the median, the box edges show the 25th and 75th percentiles and the whiskers show the smallest and highest value within 1.5 box lengths from the box (*n* = 8 for 35 W Sedentary; *n* = 9 for 35 W Exercised, 61 W Sedentary and 61 W Exercised). * *p* < 0.05; ** *p* < 0.01. a—Significantly different from 35 W Sedentary; b—significantly different from 61 W Sedentary; c—significantly different from 35 W Exercised.

**Table 1 ijms-23-11619-t001:** Effect of age and exercise training on animals’ body weight and testes–body weight ratios, and on circulating testosterone. Results are expressed as the mean ± SEM (*n* = 8 for 35 W Sedentary; *n* = 9 for 35 W Exercised, 61 W Sedentary and 61 W Exercised). * *p* < 0.05; ** *p* < 0.01. a—Significantly different from 35 W Sedentary; b—significantly different from 61 W Sedentary; c—significantly different from 35 W Exercised.

Weight	35 W Sedentary	35 W Exercised	61 W Sedentary	61 W Exercised
Animal (g)	471.728 ± 9.820	404.174 ± 4.570 a **	543.520 ± 4.420 a **	443.401 ± 6.649 b **, c **
Testis (mg)	1642.875 ± 23.112	982.300 ± 48.160 a **	1525.000 ± 32.632 a **	1173.889 ± 161.089
Testis/body weight (mg/g)	3.447 ± 0.058	2.535 ± 0.170 a **	2.831 ± 0.079 a **	2.693 ± 0.376
Serum testosterone levels (pg/mL)	111.624 ± 24.412	56.2395 ± 48.690	128.428 ± 72.238	257.000 ± 99.458 b *, c *

**Table 2 ijms-23-11619-t002:** Influence of exercise training on rat’s sperm concentration and morphology. The results are expressed as the mean ± SEM. * *p* < 0.05; ** *p* < 0.01. N, normal; DH, decapitated head; FH, flattened head; PH, pin head; BN, bent neck; TD, tail defect.

Parameter	61 W Sedentary	61 W Exercised
Sperm concentration (×10^6^/mL)	23.111 ± 1.817	18.460 ± 1.738
Sperm morphology		
Normal (%)	72.480 ± 2.360	43.650 ± 3.780 **
DH (%)	6.700 ± 0.571	15.550 ± 4.520 *
FH (%)	1.616 ± 0.233	0.926 ± 0.166
PH (%)	0.067 ± 0.0410	0.030 ± 0.028
BN (%)	8.542 ± 1.930	6.845 ± 1.739
TD (%)	10.630 ± 1.380	33.630 ± 3.690 **

## Data Availability

The data presented in this study are available in the present article and Appendix A.

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
