# Peer review of "Effects of Age and Lifelong Moderate-Intensity Exercise Training on Rats’ Testicular Function"

_ijms, 2022, doi:10.3390/ijms231911619_

Round 1

Reviewer 1 Report

This study of rat testical changes related to age and exercise is professionally presented. A few suggestions are provided in the attached pdf file.

Author Response

The authors really appreciate your comments. All abbreviations were verified and are now fully defined. We did not include a list of abbreviations since the template provided does not have this section. However, we would be happy to include it if the editor considers it relevant. Additionally, the reference list was updated as suggested by the reviewer. Some older articles are usually the first reports or the most relevant findings for the subject presented and thus, the citations were maintained in these cases.

Reviewer 2 Report

The manuscript submitted by Joana V. Silva and colleagues describes the impact of moderate-intensity exercise training on rat’s testicular function. Exercise surprisingly decreases Nrf1 and other mitochondrial biogenesis biomarkers mRNA levels along with lower levels of OXPHOS complexes subunits. In addition, exercise leads to higher levels of oxidative stress and reduced translation. Although the study is exciting and the results are clearly exposed, it is weak in some aspects. Testes are cell heterogeneous, and it is never characterized the differences in the distinct cell types by immunohistochemistry along the manuscript. It is feasible that exercise only impacts one stage of spermatogenic differentiation, particularly considering the divergent dependence on mitochondrial activity. Thus, I cannot accept the manuscript for publication without addressing this point. Other critical questions are indicated below.

 1-     Why did you look SOD1 and GPX4, and no other antioxidant proteins? For example, considering your results, I suggest looking at mitochondrial SOD2, and not only the cytosolic SOD1. Please, justify with references or include a high panel of antioxidant enzymes

2-     I would avoid talking about translation only looking at some proteins like mTORC1 or eIF2α. I suggest discussing but not concluding it if you do not have some measures of protein synthesis. In addition, please say “translation” or “protein synthesis”, but “protein translation” is not scientifically correct.

3-     Since the results after training are convincing but suppressing, did you measure the circulating testosterone levels in the different groups? It will be supportive of the discussion.

4-     Figure 2a-c: It would be beneficial to have the protein levels of Nrf1, Pgc1α, and Sirt1. Please, use references to explain why you used these markers to represent mitochondria dynamics.

5-     Please, add representative western blots in the main figures and not only as supplementary figures.

Author Response

We thank the reviewer for the valuable comments that we believe have contributed to improve the manuscript. Please find our point-by-point response below, detailing the revisions in the manuscript and our responses to the reviewers' comments. We used track changes to highlight the changes made to the manuscript.

The manuscript submitted by Joana V. Silva and colleagues describes the impact of moderate-intensity exercise training on rat’s testicular function. Exercise surprisingly decreases Nrf1 and other mitochondrial biogenesis biomarkers mRNA levels along with lower levels of OXPHOS complexes subunits. In addition, exercise leads to higher levels of oxidative stress and reduced translation. Although the study is exciting and the results are clearly exposed, it is weak in some aspects. Testes are cell heterogeneous, and it is never characterized the differences in the distinct cell types by immunohistochemistry along the manuscript. It is feasible that exercise only impacts one stage of spermatogenic differentiation, particularly considering the divergent dependence on mitochondrial activity. Thus, I cannot accept the manuscript for publication without addressing this point.

Reply: The reviewer touched on a very pertinent aspect, which should perhaps be one of the main weaknesses of the presented study. The evaluation of the contribution of whole cell types will possibly have a diluting effect on the contribution of some molecular pathways for testis remodelling induced by aging and exercise as, for example, metabolic ones since the reliance on oxidative metabolism, and consequently, on mitochondria is distinct among testes cell types, as pointed out by the reviewer. IHC is a powerful approach to characterize distinct cell types in tissue samples, which should be combined with the molecular analysis of isolated cells to evaluate the contribution of each cell type adaptation to whole testes remodelling, which justifies another study. Also, we do not have enough tissue available to characterize the differences among the different cell types by immunohistochemistry, so we cannot do the suggested. Nevertheless, this limitation has been added and discussed in the Discussion section, in particular on page 7, lines 229-235. The reviewer can now find the following: “When interpreting these results, it is important to consider that testes are a highly heterogenous type of tissue, composed of many types of cells (Sertoli and Leydig cells, germ cells – spermatogonia, spermatocyte, spermatid, spermatozoa, myoid cells, etc.), with different dependence on mitochondria activity. Therefore, it will be important in the future to characterize the different types of cells both in terms of mitochondrial biogenesis and antioxidant profile, to further deepen the evidence here presented.”

Other critical questions are indicated below.

1-     Why did you look SOD1 and GPX4, and no other antioxidant proteins? For example, considering your results, I suggest looking at mitochondrial SOD2, and not only the cytosolic SOD1. Please, justify with references or include a high panel of antioxidant enzymes

Reply: This is a relevant observation. It is known that aging was associated with increased oxidative stress and free radicals production, and it was already shown that mitochondrial dysfunction is involved in testicular aging. This increase is especially due to alterations in the enzymatic activity of anti-oxidant molecules such as glutathione peroxidase (GPx) and superoxide dismutase (SOD)1–4. Despite GPx4 being expressed in most mammalian tissues, larger amounts are present in testes and spermatozoa, including in mitochondria. Also, since the levels of OXPHOS complexes were diminished, it was expected that their activity also decreases, thus, an increase in ROS with mitochondrial origin is not expected. Consider that, we choose to look to SOD1 since it is ubiquitously expressed with predominant localization in the cytoplasm, but it is also localized within mitochondria, nucleus, and the ER, and constitutes the first line of defence against hypoxia-induced oxidative stress and superoxide radicals. We include this information in the discussion (Page 7, lines 260-269).

2-     I would avoid talking about translation only looking at some proteins like mTORC1 or eIF2α. I suggest discussing but not concluding it if you do not have some measures of protein synthesis. In addition, please say “translation” or “protein synthesis”, but “protein translation” is not scientifically correct.

Reply: We appreciate the reviewer’s corrections. The terms were corrected throughout the manuscript.

3-     Since the results after training are convincing but suppressing, did you measure the circulating testosterone levels in the different groups? It will be supportive of the discussion.

Reply: We appreciate your suggestion. The levels of circulating testosterone in each group were included in Table 1 (page 2) and described in section 2.1. of the results (Page 3, lines 94-96). The methodology used was described in the materials and methods section (Page 10). The discussion of the results was integrated into the discussion (page 7, lines 240-246), as suggested.

4-     Figure 2a-c: It would be beneficial to have the protein levels of Nrf1, Pgc1α, and Sirt1. Please, use references to explain why you used these markers to represent mitochondria dynamics.

Reply: We choose to measure mRNA because there are several studies that evaluate the effect of exercise in other tissues by measuring Pgc1alpha mRNA (doi: 10.1016/s0300-9084(99)00223-0) and considering that we do not have antibodies for Nrf1, Pgc1α, and Sirt1available. We choose these markers to represent mitochondria dynamics since Pgc1a is the master player of mitochondrial biogenesis, controlling the expression of genes involved in energy homeostasis, mitochondrial biogenesis, fatty acid oxidation and glucose metabolism (https://doi.org/10.1371/journal.pone.0011707). SIRT1 converts inactivated PGC-1α to the active form and active Pgc1a stimulates the expression of Nrf1 and Nrf2 that act on the nuclear genes coding for subunits of the OXPHOS system (https://doi.org/10.1016/j.bbabio.2013.12.008; 10.1093/jpp/rgac011). PGC-1α is pivotal for mitochondrial function, as well as for the expression of key mitochondrial proteins, such as Sirtuin 3. Also, Nrf1 and Sirt1 seems to regulate testosterone biosynthesis (doi: 10.1016/j.jsbmb.2019.04.019; 10.1007/s13238-020-00771-1; 10.1016/j.fertnstert.2012.04.008), which could partially explain some of the alterations in testosterone levels observed. Data obtained for Nrf1, Pgc1α, and Sirt1 transcripts are in accordance with OXPHOS complexes subunits levels. Overall, data obtained do not support the effect of endurance exercise in enhancing PGC1alpha expression reported in other tissues such as skeletal muscle (doi: 10.1016/s0300-9084(99)00223-0). Some of these references were added to the manuscript on lines 126-130 and lines 258-259.

5-     Please, add representative western blots in the main figures and not only as supplementary figures.

Reply: The representative western blots were included in figure 2 as suggested by the reviewer.

Round 2

Reviewer 2 Report

I thank the authours to answer all the questions and address some of them. The increase of circulating testosterone levels in exercised mice is neeat and fascinating. I recommend discussing these results in relation wih mitochondrial activity and redox balance. In addition, I miss the GAPDH blot in figure 2, as well as the representative blots in Figure 3. Why not all the proteins from the ETC antibody cocktail are detected?

In addition, I am still not convinced with the point 1, and I strongly suggest evaluating the levels of the different antioxidant enzymes. Regarding the bibliography, you can find the important activity of SOD2 or catalase in testes. 

Author Response

We appreciate the reviewer's comments which helped us to improve the manuscript. Please find our point-by-point responses to the reviewers' comments below. We used track changes to highlight the changes made to the manuscript.

I thank the authors to answer all the questions and address some of them. The increase of circulating testosterone levels in exercised mice is neat and fascinating. I recommend discussing these results in relation with mitochondrial activity and redox balance.

Reply: This is a very pertinent suggestion. It has been described that the decrease in testosterone production by Leydig cells in aged models can be explained by the imbalance of pro-oxidants and antioxidant defence system, especially by reduced levels of SOD and GPx activities (doi: 10.2164/jandrol.05075). In this work we did not find significant alterations in testosterone levels with aging, except in exercised older rats compared to the younger group where the levels of testosterone increased, also accompanied by an increase in antioxidant enzymes (SOD1, SOD2 and GPx4).  We also showed that exercise induced beneficial mitochondrial adaptations especially in old animals including an increase in mitochondrial antioxidant capacity (SOD2), despite the decrease in mitochondrial biogenesis and activity. This is in part consistent with other studies that reported a suppression in the redox system and, thus, mitochondria-mediated ROS overload after high-intensity continuous running (doi: 10.1111/and.14520). Testosterone seems to have important mito-protective and anti-mitophagy functions in many tissues (doi: 10.1530/JOE-14-0638; 10.1089/neu.2018.6266; 10.1016/j.mce.2019.110631; 10.1007/s13353-020-00550-y), alleviating mitochondrial ROS accumulation (doi: 10.1016/j.abb.2018.05.002) and inducing the expression of Nrf1 (10.1016/j.mce.2019.110631), which was not observed in this study. Despite quite contradictory, these results may suggest a possible mechanism of protection in testis, in which age and physical exercise reduced mitochondrial activity but stimulate the antioxidant defence system preventing severe testicular damage. This information was added to the discussion on lines 312-314 and 320-332.

In addition, I miss the GAPDH blot in figure 2, as well as the representative blots in Figure 3. Why not all the proteins from the ETC antibody cocktail are detected?

Reply: We add the representative blots to figure 3 and the GAPDH blot in figure 2, as requested by the reviewer. Concerning the results obtained for the quantification of the proteins from the ETC, we used a cocktail of antibodies that detects an epitope of 5 different peptides of each ETC protein. These peptides are extremely temperature labile, and we had to make a compromise between the extraction of the proteins from the membrane (and their denaturation in order to expose the epitope) and their integrity. In our tissue samples, the best conditions allowed us to visualize four of the targeted proteins and evaluate their levels. Due to its particular lability, we could not detect the peptide of complex IV in our samples after extraction and denaturation. Still, a vast majority of the alterations that have been described in the functionality of mitochondria associated with alterations in the expression levels of ETC complexes, have been linked to CI and/or CIII and we decided to focus on those complexes.

In addition, I am still not convinced with the point 1, and I strongly suggest evaluating the levels of the different antioxidant enzymes. Regarding the bibliography, you can find the important activity of SOD2 or catalase in testes.

Reply: Thank you for your comment. We are aware of the importance of SOD2 and Catalase in testis. We evaluate the levels of SOD2 in our samples as suggested by the reviewer and the results were included in Figure 3e, Table S1 and lines 190-192 and 227-228 of results. Also, these results were integrated and discussed in Discussion section.   

Round 3

Reviewer 2 Report

I thank the authors for their detailed response. I will accept the manuscript for publication, but before I suggest including why you could not detect all the peptides from the ETC cocktail in the Material and Methods. It is interesting for other researchers.